# A protocol for enhancing the diagnostic accuracy and predictive validity of neonatal opioid withdrawal syndrome: The utility of non-invasive clinical markers

Sarah E. Maylott[1]*, Barry M. Lester[2,3,4,5], Lydia Brown[6], Ayla J. Castano[6], Lynne Dansereau[2], Sheila E. Crowell[7], Pascal Deboeck[2], Amy Salisbury[5,6,8], Elisabeth Conradt[1]

1 Department of Psychiatry & Behavioral Sciences, Duke University, Durham, NC, United States of America, 2 Center for the Study of Children at Risk, Alpert Medical School of Brown University, Providence, Rhode Island, United States of America, 3 Department of Psychiatry, Alpert Medical School of Brown University, Providence, Rhode Island, United States of America, 4 Department of Pediatrics, Alpert Medical School of Brown University, Providence, Rhode Island, United States of America, 5 Women and Infants Hospital, Providence, Rhode Island, United States of America, 6 Department of Psychology, University of Utah, Salt Lake City, Utah, United States of America, 7 Department of Psychology, University of Oregon, Eugene, Oregon, United States of America, 8 School of Nursing, Virginia Commonwealth University, Richmond, Virginia, United States of America

* sarah.maylott@duke.edu

**Data Availability Statement:** Deidentified research data will be made available upon reasonable

## Abstract

Every 15 minutes in the US, an infant exposed to opioids is born. Approximately 50% of these newborns will develop Neonatal Opioid Withdrawal Syndrome (NOWS) within 5 days of birth. It is not known which infants will develop NOWS, therefore, the current hospital standard-of-care dictates a 96-hour observational hold. Understanding which infants will develop NOWS soon after birth could reduce hospital stays for infants who do not develop NOWS and decrease burdens on hospitals and clinicians. We propose noninvasive clinical indicators of NOWS, including newborn neurobehavior, autonomic biomarkers, prenatal substance exposures, and socioeconomic environments. The goals of this protocol are to use these indicators shortly after birth to differentiate newborns who will be diagnosed with NOWS from those who will have mild/no withdrawal, to determine if the indicators predict development at 6 and 18 months of age, and to increase NOWS diagnostic sensitivity for earlier, more accurate diagnoses.

## Introduction

Substance use during pregnancy is a public health crisis. Approximately 14–22% of pregnant women fill a prescription for opioids, and close to 7% report using opioids while pregnant [1]. About half of newborns exposed to opioids *in utero* are diagnosed with Neonatal Opioid Withdrawal Syndrome (NOWS) [2]. These newborns become physiologically dependent while *in utero* and endure drug withdrawal following the cessation of exposure after delivery.

request when the study is completed and published

**Funding:** This work is funded by the National Institute of Drug Abuse [R01DA049755 to BL and EC]. The funders did not and will not have a role in the study design, data collection and analysis, decision to publish, or preparation of the manuscript. https://nida.nih.gov/.

**Competing interests:** The authors have declared that no competing interests exist.

Hospitalization of infants diagnosed with NOWS results in around $1.5 billion in hospital charges across a decade [3], and these infants are four times more likely to require post discharge rehospitalization than children without a NOWS diagnosis [4]. The impact of NOWS lingers beyond birth, with developmental [5] and educational deficits [6]. Presently, there are no evidence-based clinical markers of NOWS [7]. The absence of reliable and valid diagnostic measures may limit medical professionals' ability to provide optimal care to opioid exposed newborns. We propose research-based clinical markers for identifying infants with NOWS.

## Neonatal opioid withdrawal

The pathophysiology of opioid withdrawal is complex in neonates, given their rapid neurological development [8]. Opioids readily cross from the placenta into fetal circulation, with increasing transmission over gestation as the placental barrier expands and thins, enhancing blood flow to the fetus [9]. Generally, opioid exposure triggers μ-, κ-, and δ-receptors concentrated in the central nervous system and gastrointestinal tract [10]. In neonates, while δ- and κ-receptors are less developed, their μ-receptors' density and affinity are comparable to adults [11]. Consequently, the μ-receptor binding of opioids and then the cessation binding (i.e., the cessation of opioid exposure) after birth may lead to withdrawal signs, including autonomic instability, high-pitched cry, irritability, tremors, and difficulty feeding and sleeping [12].

## Challenges and potential advancements to NOWS diagnostics and clinical care

### Restrictions to current diagnostic guidelines for NOWS

To guide treatment of NOWS, clinicians use observational diagnostic tools to evaluate newborn central nervous system disturbances, metabolic, vasomotor, and respiratory disturbances, and gastrointestinal disturbances [13, 14]. However, these diagnostic tools are not often thoroughly researched, psychometrically sound, and/or standardized, and the choice of diagnostic tool depends on independent hospital protocols [15]. Many criteria for diagnosis are subjective, with scoring varying not only between hospitals but also between hospital staff [7]. Unfortunately, clinicians do not have a predictive model to determine NOWS onset or severity. With the current tools, most neonates are not diagnosed with NOWS until 36 hours after birth and withdrawal signs can be delayed 5 to 7 days, depending on the opioid's half-life [10], delaying optimal care. This "wait-and-see" approach may be unnecessary for the 20–40% of newborns who will not develop NOWS. Furthermore, the dichotomous diagnosis of NOWS ignores the variability in NOWS severity and limits the predictive validity.

### Proposed advancements for NOWS diagnostic protocols

In this protocol we propose several biobehavioral markers of NOWS including the NeoNatal Neurobehavioral Scale (NNNS-II), newborn cry and sleep, heart rate variability (HRV), salivary cortisol, and epigenetic markers to advance diagnostic procedures for NOWS. We suggest that these markers will advance the current clinical protocols by 1) predicting NOWS onset and severity, 2) developing a protocol to detect risk for NOWS 24 hours after birth, 3) measuring NOWS severity outcomes as a spectrum influenced by exposure type, frequency, and co-substance use, 4) using the spectrum of NOWS severity to predict longitudinal neurodevelopment 5) and as an indicator of successful treatment (Table 1).

**Table 1. Advancing the science of NOWS research.**

| Current clinical and research practices | Proposed advancement |
|---|---|
| Not predictive of NOWS onset or severity | Establishing evidence-based clinical markers that predict onset and severity of NOWS |
| NOWS is often not diagnosed until 36 hours after birth and can be delayed 5 to 7 days | These predictors could be used to develop an algorithm that detects risk for NOWS 24 hours after birth |
| Utilizes a simple binary outcome in diagnosis (i.e., yes/no) | Measure NOWS severity as a spectrum of outcomes influenced by type of exposure, frequency of use, & other substance exposure |
| Predicts probability of adverse neurodevelopmental outcomes from a dichotomous diagnosis | Use evidence-based clinical markers to predict longitudinal neurodevelopment outcome by a continuous distribution of NOWS severity |
| Measures length of hospital stay as an indicator of a successful treatment outcome without considering other factors that impact length of stay, such as CPS (i.e., foster care) involvement | Measure NOWS severity as an indicator of successful treatment outcome |

## Clinical measure 1: Newborn neurobehavior

The NNNS-II is a standardized assessment of both newborn neurologic integrity and behavioral functioning based on the original NNNS exam [16], which captures differences in exposed newborns who require pharmacological treatment from those who do not need intervention [17]. Moreover, newborn neurobehavior predicted the resolution of NOWS, suggesting that the NNNS-II can be used to identify newborns at risk for NOWS [18]. The original NNNS was also predictive of developmental outcomes through 4.5-years of age children, including behavior problems, school readiness, and IQ [19].

## Clinical measure 2: Newborn cry acoustics

We can further increase objectivity of NOWS assessments through automated measures. "Abnormal" or "high-pitched" newborn crying is a primary NOWS sign, and yet, is one of the most subjective and poorly quantified [7]. Despite scoring limitations, crying is a crucial biobehavioral marker for NOWS diagnoses. *In utero* substance exposure increases central nervous system irritability, activates the hypothalamic-limbic system, and triggers abnormal infant crying [20]. Recording and analyzing infant crying offers a way to accurately capture and interpret atypical cry characteristics. Cry analysis relies on specific mechanisms in cry production, including threshold and latency, indicative of central nervous system reactivity, energy, dysphonation, and utterances, signifying respiratory control, and fundamental frequency, hyperphonation, formant frequencies, and mode changes, demonstrating neural control of the vocal tract [21]. Infants with *in utero* opioid exposure tend to have shorter cry utterances and extremely high-pitched cries, and infants co-exposed to opioids and cocaine have comparably louder and higher pitched cries [21]. High-pitched and hyperphonated cries have been reported in infants with neurologic problems [22], suggesting that infants with substance exposures, especially those with polysubstance exposures may be at increased neurological risk. Preliminary acoustic cry data showed that it was 81% accurate in predicting NOWS; moreover, acoustics were more accurate in some cases when the observational crying score was incorrectly evaluated (Unpublished data). Abnormal infant crying has also been linked to delays in cognitive and motor functioning at 24 months of age [22]. This automated assessment could help identify infants at greater risk for NOWS and subsequent neurodevelopmental disruptions, while also shortening assessment times, reducing burdens on clinical staff, and limiting subjectivity of observational judgements.

## Clinical measure 3: Newborn sleep patterns

Disrupted sleep is another important indicator of NOWS, as a sign of nervous system dysregulation [23] or interruptions of autonomic regulation of arousal and movement during sleep [24]. Infants exposed to opioids *in utero* displayed greater wakefulness with disruptions in the amount of REM and non-REM sleep, suggesting sleep state instability [25]. Further, newborns exposed to higher doses of opioids *in utero* showed more disturbances in neonatal sleep, indicating that sleep patterns provide insight into the neurobehavior of newborn withdrawal [26]. Using sleep mattress recordings, we categorize disrupted sleep patterns associated with NOWS [12]. We propose to utilize these automated tools to help limit diagnostic subjectivity, improve diagnostic sensitivity, ease clinician burdens, and reduce healthcare costs [27].

## Clinical measure 4: Newborn heart rate variability

Like the biobehavioral measures, biological markers, such as newborn heart rate variability (HRV), salivary cortisol, and epigenetic mechanisms also provide vital information for identifying newborns at higher risk for NOWS. HRV—the beat-by-beat fluctuations in cardiovascular activity—reflects newborns' autonomic nervous system functioning, which controls involuntary physiological responses through the sympathetic and parasympathetic systems [28]. These systems work in opposition to create homeostasis; the sympathetic nervous system works to increase heart rate while the parasympathetic nervous system decreases heart rate [28]. Fetal HRV is often suppressed in opioid exposed infants, especially those exposed to more than one substance, reflecting a dysfunction in this balance [29, 30]. Furthermore, newborns prenatally exposed to opioids had higher heart rate, lower HRV, and more blunted respiratory sinus arrythmia—beat-to-beat variability in heart rate that occurs with respiration —than non-exposed newborns [31]. HRV was also less variable in prenatally exposed newborns diagnosed with NOWS compared to those exposed newborns not diagnosed [32]. Newborn HRV could help identify those infants at greater risk for developing severe withdrawal signs.

## Clinical measure 5: Newborn salivary cortisol

Salivary cortisol is another non-invasive biological indicator that may help identify newborns at higher risk for severe NOWS signs, facilitating the detection of newborns' dysregulated stress responses. Prenatal opioid use is often associated with elevated life stressors such as poverty, psychopathology, and poor general health, as well as exposure to violence, trauma, and/or abuse. Maternal stress can negatively impact fetal development [33]. This process can happen through a variety of mechanisms, including via heightened maternal cortisol levels in the bloodstream, through the reduced expression of placental 11 beta-hydroxysteroid dehydrogenase (11 beta-HSD), which converts active cortisol to inactive cortisone [34], and by autonomic nervous system activation, which may diminish blood flow to the placenta, limiting fetal nutrients and oxygen [35]. Excessive exposure to maternal stress hormones during pregnancy is theorized to adversely impact fetal programming, adapting fetal physiology to the expected postnatal environment by modifying their stress response system's (hypothalamic-pituitary-adrenal axis; HPA axis) set points [36, 37]. *In utero* exposure to opioids may also contribute to this fetal programming [36]. In animal models, prolonged *in utero* exposure to opioids suppressed opioid receptors, leading to offspring with impaired or overactive negative feedback regulation of their HPA axis [38]. In fact, newborn salivary cortisol remains elevated in infants who develop more severe withdrawal, potentially indicating HPA axis dysregulation or an altered stress response due to the severity of withdrawal. Regardless of the

pathophysiology, salivary cortisol offers a measure of newborn neuroendocrine functioning that is reflective of risk for NOWS [39].

## Clinical measure 6: Newborn epigenetics

Epigenetic mechanisms offer insight into the variability of NOWS diagnoses and severity. One such mechanism is DNA methylation, a process by which a methyl group attaches to a particular region on the DNA, effectively preventing transcription and reducing the likelihood that the gene will be expressed [40]. Stress response genes (*COMT*, *NR3C1*), opioid receptor genes (*OPRM1*, *OPRK1*, *PNOC*), and opioid-related genes (*ABCB1*, *CYP2D6*) are associated with newborn withdrawal from prenatal opioid exposure and may be useful markers of susceptibility to NOWS and pharmacotherapy [41, 42]. For example, the *COMT* gene has been associated with shorter hospital stays for neonates prenatally exposed to opioids. This gene is associated with the dopaminergic system and expression of this gene has been linked to increases in circulating catecholamines—a neurotransmitter that helps the body respond to stress—potentially improving stress tolerance in opioid-exposed infants [43]. Prenatal DNA methylation of *NR3C1*—a gene associated with glucocorticoid receptors—has been linked to greater cortisol reactivity in infancy [44], which may indicate NOWS severity. Methylation of this gene likely reduces cortisol binding sites by limiting the number of glucocorticoid receptors, increasing the amount of cortisol in the bloodstream. Along with neonatal salivary measures of cortisol, identifying the epigenetic mechanisms related to adverse prenatal environments may help pinpoint infants prenatally exposed to opioids with altered stress response systems, who are more likely to need pharmacological interventions for NOWS. Furthermore, prenatal opioid exposure is related to increased DNA methylation in opioid receptor and opioid-related genes [41, 42]. Higher levels of *OPRM1* promoter methylation at transcription factor binding sites have been associated with more severe NOWS signs and an increased need for pharmacological intervention [45, 46]. This hypermethylation is theorized to downregulate *OPRM1* gene expression and reduce μ-opioid receptors, resulting in neonates needing more medication to combat withdrawal [46]. Epigenetics could be a vital clinical marker of newborns' withdrawal severity and response to pharmacotherapy [47].

## Clinical measure 7: Maternal toxicology

Use of maternal hair for prenatal substance use toxicology is another key measure, providing information on the levels and types of opioid exposure and other drug metabolites across pregnancy [48]. We can determine potential risk and severity for neonatal withdrawal depending on type, timing, or amount of exposure [49]. Newborns exposed to higher doses of methadone *in utero* were more likely to be treated for NOWS and to spend longer in the hospital than newborns exposed to lower doses [50]; however, the effect of opioid dose on the severity of withdrawal continues to be debated. Further, newborns were treated for NOWS longer if they had a longer gestation (37–42 weeks treated for an average of 39.5 days; 33–36 weeks treated for an average of 20 days; 23–32 weeks treated for an average of 10 days). These findings may be due to the high permeability of the placenta in the late third trimester [51] or that current assessments are not geared toward assessing NOWS in preterm infants who may display differences in neurological development and, therefore, withdrawal signs at birth.

Identification and application of these evidence-based predictors of NOWS could advance clinical care of newborns with prenatal opioid and other substance exposures, leading to a standardized screening routine for all newborns at risk of developing NOWS, and even flag newborns not initially identified as opioid-exposed.

## Developmental consequences of prenatal opioid exposure

The implications of *in utero* opioid exposure extend beyond birth [52], with exposed infants displaying lower cognitive functioning than their non-exposed peers, a divergence that appears magnified as children age [53, 54]. These children show cognitive and language impairments around 2 months [55], inhibitory control difficulties at 2 years [56], and externalizing and internalizing behaviors, social issues, attention problems [54], and lower executive functioning at 4.5 years of age [57]. Extant literature focuses on developmental differences in those infants exposed to opioids *in utero* and non-exposed infants. However, differences between exposed infants diagnosed with NOWS and exposed infants *not* diagnosed with NOWS could highlight important risk and resiliency factors [58]. Furthermore, few studies consider adverse socioeconomic circumstances, maternal psychopathology, or polysubstance use [59], which may impact both prenatal and postnatal environments [36]. The postnatal environment plays a substantial role in infant development, and yet, is understudied. However, large-scale studies, such as the Maternal Lifestyle Study and The Infant Development, Environment, and Lifestyle (IDEAL) Study have found that early adversity is a strong determinant of child problem behavior, partially explaining the relationship between prenatal substance exposure and developmental outcomes [60, 61].

## Current study

Our study protocol has three aims. First, to determine NOWS onset and severity using a battery of sensitive clinical markers (Fig 1). Second, to evaluate the predictive validity of these clinical markers on neurodevelopmental outcomes at 6 and 18 months and test whether NOWS severity predicts neurodevelopment impairment. Third, to increase NOWS clinical diagnostic sensitivity and objectivity for earlier, more accurate diagnoses.

## Methods

This study was approved by the Institutional Review Board at the University of Utah on September 14[th], 2019 (00124221) and Women and Infants Hospital on February 13[th], 2020 (1479081–3). Written consent was obtained for all participants. Enrollment and data collection began in 2019 and will be completed in 2025.

## Study population

We collect data from 312 newborns with prenatal opioid and other substance exposure across two sites, Women and Infants Hospital of Rhode Island and the University of Utah Hospital. We collect follow-up data at 6 and 18 months of age. Both sites recruit participants prenatally or shortly after birth (Table B in S1 Text). Birthing parents are over 18 years old and identified to have used opioids during pregnancy via medical records. Newborns are medically stable and include singleton and twin pregnancies, term and preterm infants, and breast- and bottle-fed infants. Newborns with congenital abnormalities, genetic syndromes, metabolic disturbances, or serious medical illnesses, such as sepsis, asphyxia, seizures, or respiratory failure are excluded. Newborns are also excluded if they are unable to take oral medications or if their caregiver is unable to provide informed consent (Table A in S1 Text).

### Newborn (Time 1)

**Procedure.** We collect data approximately 24–48 hours after birth, prior to treatment for NOWS. The birthing parent/legal guardian provides informed consent and is asked to fill out

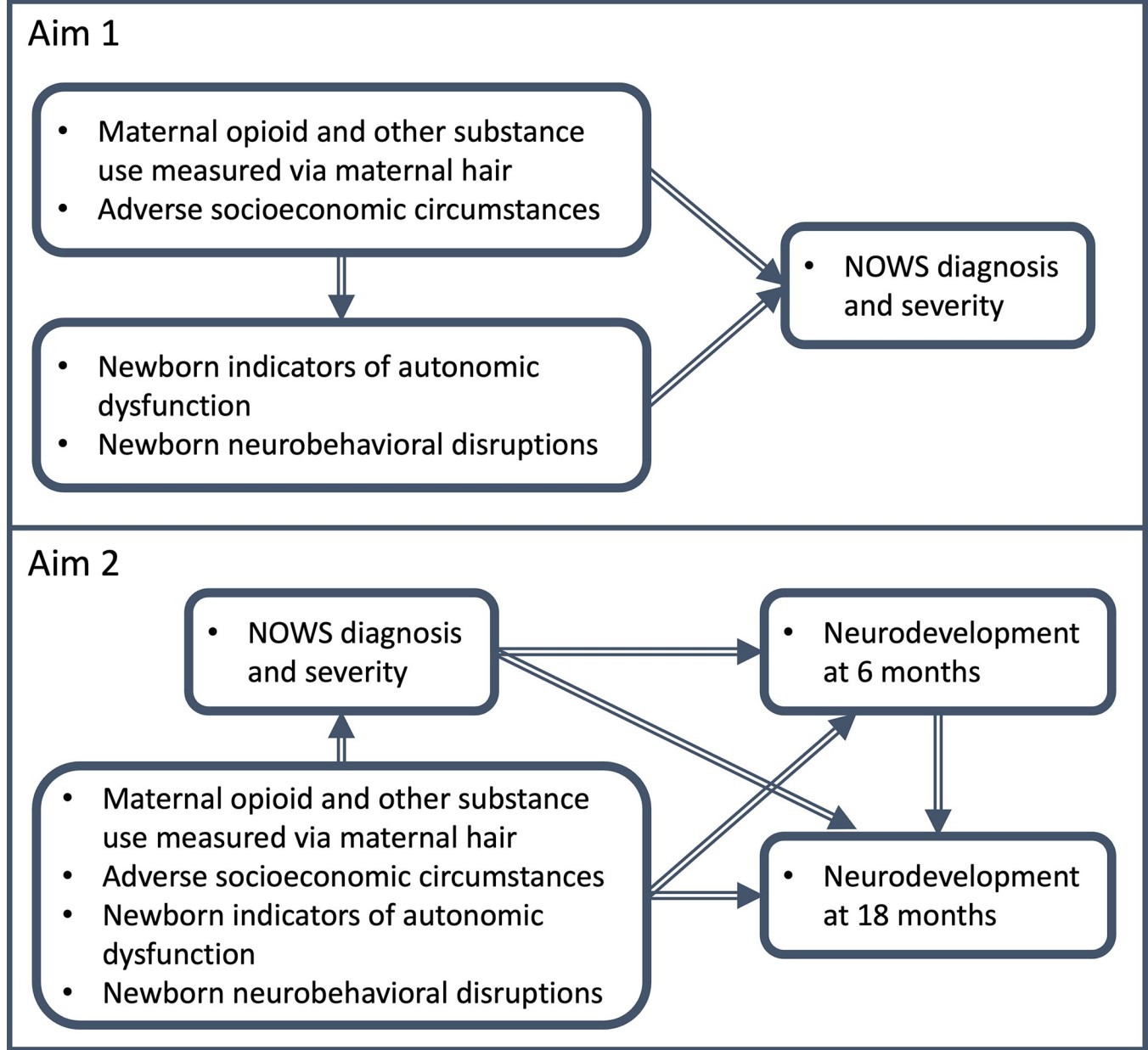

**Fig 1.** Conceptual model for Aim 1 (top) and Aim 2 (bottom).

questionnaires assessing maternal sociodemographic risk and psychopathology. Newborn birth characteristics are extracted from medical records.

For the newborn assessments, we begin by collecting the first of three saliva samples using a small cotton swab. Then we measure infant heartrate, placing three electrodes on the upper (negative) and lower (positive) chest around the infant's heart, and on the right lower side of the chest (neutral) and are left on the infant for the full newborn visit. For the next 10 minutes, infant heartrate is monitored. Next, we assess neurobehavior with the NNNS-II, which lasts about 15 minutes. The cry recording begins during the NNNS-II, is stopped after the assessment concludes, and then is restarted to record continuously for the length of hospital stay.

Caregivers and hospital staff are asked to remain quiet while newborn cries are recorded. After the NNNS-II, we measure infant heartrate again for 10 minutes. We collect the second and third saliva samples 20- and 30-minutes post-NNNS-II assessment, respectively, and the buccal sample. ECG leads and wires are removed post newborn assessments, so that we cause no additional stress during the exams. Finally, we collect maternal hair samples for toxicology analyses. If the hair is too short or birthing parents refuse to provide the hair sample, the missing data is tracked but participants are still included in the study.

We also record the number of primary caregiver changes over the course of the study, starting at birth. This occurs when there is a change in custody. The child's primary caregiver at birth is typically the biological mother. Caregiver transitions occur when the child is placed with a foster parent, adoptive parent, or other caregiver (e.g., family member). Reasons for caregiver transitions include child protective services involvement (e.g., neglect, abuse, substance use), incarceration of caregiver, death of caregiver, adoption, and/or reunification, among others. Caregiver changes may be an indicator of problem behavior [62, 63]. Next, we set up a passive sleep sensor and cry recorder on the newborn's bassinet/pram. Newborn sleep and cry are checked daily and recorded continuously throughout infants' stay in the hospital beginning with the start of the NNNS newborn assessment.

**Maternal measures.** *Maternal sociodemographic risk*. Maternal sociodemographic risk is comprised of socioeconomic status and financial insecurity. The birthing parent reports socioeconomic factors with a brief demographic survey and financial insecurity with the Family Resource Scale [64], which measures whether the family has income for basic needs, time for self, and time for family. The family resource scale has high test-retest reliability and predictive validity [64].

*Maternal psychopathology*. Maternal psychopathology is measured with the Patient Health Questionnaire-9 (PHQ-9), four Patient-Reported Outcomes and Measurement (PROMIS) surveys from the NIH Toolbox, the Difficulties in Emotion Regulation-Short Form (DERS-SF), the Borderline Symptoms Checklist, "My Inner Feelings" (BSL-23), and the Connor-Davison Resilience Scale (CD-RISC). The PHQ-9 [65] is a 9-item self-report measure of depression with good psychometric properties. The PROMIS assesses anger (5-items), anxiety (8-items), depression (8-items), and emotional support (4-items) [66]. The 18-item DERS-SF [67] is an assessment of emotion dysregulation, with strong psychometric properties. The BSL-23 is a self-rating measure of borderline symptomatology, with good psychometric properties [68]. Lastly, the Connor-Davidson Resilience Scale (CD-RISC-10) is 10 items, each rated on a 5-point scale (0–4), with higher scores reflecting greater resilience [69].

*Maternal trauma*. Maternal history of trauma is assessed from the Adverse Childhood Experiences (ACEs) [70] and the Benevolent Childhood Experiences (BCEs) [71]. We use 19-items from the ACEs measure to retrospectively assess the possibility of maltreatment in childhood and as a teenager. In contrast, the BCEs is a 10-item self-report measure of positive childhood experiences relating to perceived safety and security, positive quality of life, and interpersonal support.

*Prenatal substance exposure*. Opioids and other substance exposure are measured via confirmatory toxicology analysis from the birthing parent's hair [48] by Omega Laboratories Inc. (www.omegalabs.net). Opioids and metabolites in the blood are deposited in the growing hair, which provides a historical marker of drug exposure. Therefore, we require 9 cm of hair (third trimester: 3cm closest to the scalp, second trimester: >3–6 cm from the scalp, first trimester: >6–9 cm from the scalp). We collect a lock of hair from multiple sections of the crown area— to minimize noticeability—as close as possible to the scalp and approximately the diameter of a drinking straw (e.g., 6mm or 90–120 strands of hair). We also ascertain if hair has been treated in the last 9 months, which may affect specimen integrity.

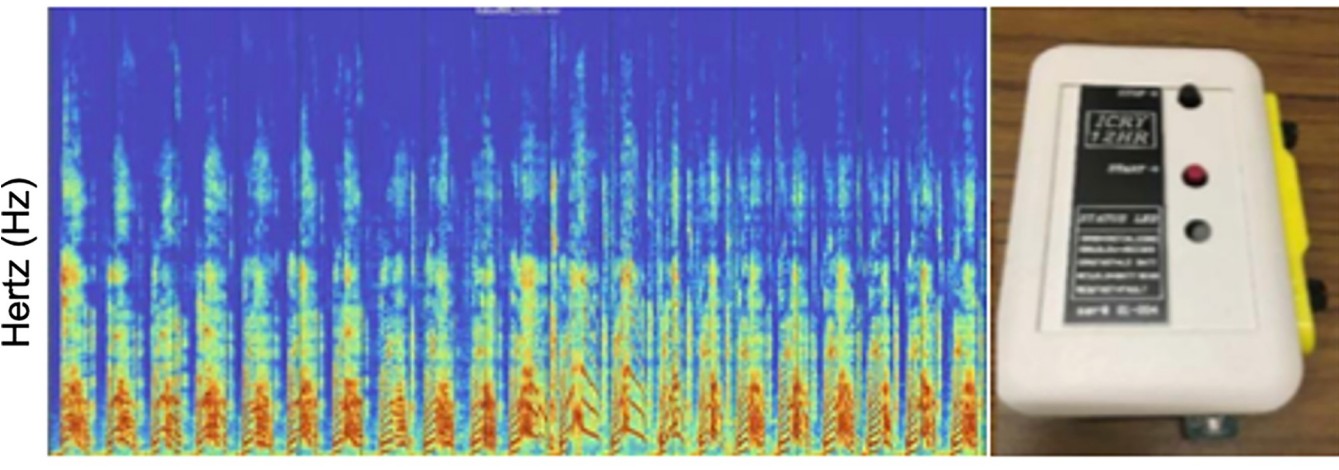

**Fig 2. Examples of digital spectrogram and cry recorder.** Red bands indicate individual cry bursts/utterances that contain pitch characteristics implicated in NOWS.

Maternal alcohol use cannot be measured via maternal hair, so it is assessed via the Substance Use Inventory (SUI; i.e., Weiss et al. [Unpublished]) [72], a self-report measure of quantity and frequency of substance use that was adapted from Timeline Follow-back interview, with high reliability for alcohol use [73]. We also abstract relevant substance use data from the medical record.

**Newborn measures.** *Neurobehavioral assessment.* The NeoNatal Neurobehavioral Scale (NNNS-II) is a standardized comprehensive evaluation of the neurobehavioral performance of newborns [16]. NNNS-II summary scores are described in Table D in S1 Text.

*Cry assessment.* The cry assessment is measured via recording of an infant's cry—a sequence of utterances and silences. An utterance is a contained vocal output, either voiced (generated via vocal vibrations) or unvoiced (due to frication or tension in the vocal tract). The infant cry recording and analysis system uses state-of-the-art Cepstral analysis (Fig 2) to extract acoustic parameters from cry recordings [74]. The cry analysis system classifies utterances or silence (time between utterances). Acoustic parameters are calculated for each sound segment (Table 2). The cry analysis system is unique from other speech analyzers since it applies

**Table 2. Cry variables.**

| Category | Output |
|---|---|
| Timing Variables | Measures of utterance duration, categorization of long & short utterances (+/-500 milliseconds), latency to cry, and inter-utterance intervals |
| Energy Variables | Measures of loudness of cry overall and within 6 frequency bands, including mean, min/max, & SD |
| Pitch Measures | Mean, min/max, SD, and amplitude of $F_0$ for each utterance, and output modeling of the shape and complexity of pitch contours across the utterance |
| Hyper-Pitch | Mean, min/max, SD, and amplitude of $F_0$ detected above 1kHertz |
| Formants ($F_1$; $F_2$) | Description of frequency centered at the first resonance of $F_0$ (mean, min/max, SD, amplitude) |
| Voicing | Number of times that reflexes on one side of the infant's body are stronger/weaker relative to the other side |
| Fricatives | Number of observed infant stress/abstinence signs across organ systems |

current digital signal processing techniques and is specifically tailored to newborn acoustic data which are sensitive to the developing vocal tract and oral cavity.

*Sleep.* We assess newborn sleep in the hospital with the Emfit QS sleep recordings from the Emfit passive under-mattress sensor (https://emfit.com/). The Emfit measures total recovery, integrated recovery, whole night heart-rate-variability RMSSD, heart and breathing rates, sleep score, sleep time, nervous system balance, the three sleep classes classification including REM, resting heart rate, tossing & turning, movement activity.

*Heartrate.* Newborn heartrate is monitored with a MindWare Technologies Mobile Impedance Cardiograph (https://mindwaretech.com/) and a Surface Pro tablet (https://www.microsoft.com/en-us/surface) with Biolab software.

*Cortisol.* Saliva is collected with SalivaBio Infant's Swabs from Salimetrics to measure infant cortisol levels and stored in Swab Storage Tubes from Salimetrics (https://salimetrics.com/collection-method/infant-swab-device/) in a -80˚C freezer until saliva is shipped to the University of Trier, Germany to be assayed.

*Epigenetics.* Buccal cell swabs are collected with an Isohelix SK-1S/MS-01 Buccal Swabs cotton swab on the inside of the infant's cheek to examine epigenetics. Buccal cell samples are stored in a -80˚C freezer until they are ready to be shipped to Emory University, Georgia for processing.

*NOWS diagnosis and severity.* NOWS diagnosis is assessed using the Modified Finnegan Neonatal Abstinence Scoring Tool [12], the Neonatal Withdrawal Inventory (NWI) [75] or the Eat, Sleep, Console (ESC) [76]. The diagnosis of NOWS is made when the Finnegan or NWI score reach a predefined threshold.

NOWS severity is measured using the length of NOWS treatment, maximum dose of treatment medication, and necessity for additional medications to treat NOWS, such as phenobarbital.

## 6 months (Time 2)

**Procedure.** For the 6-month visit, we ask the primary caregiver to complete a series of questionnaires assessing maternal sociodemographic risk, psychopathology, and trauma, as well as infant crying, sleeping, feeding, and temperament measures. Next, we assess fine and gross motor development with the Peabody Developmental Motor Scales. We also obtain childcare information and medical history from the medical records to document the number and severity of medical illnesses. Finally, we collect a buccal sample and standard pediatric anthropometry (i.e., weight, length, head circumference).

**Maternal measures.** Maternal sociodemographic risk, psychopathology, and trauma surveys are repeated from the birth visit (see Newborn Maternal Measures). If the birthing parent is not available, the primary caregiver reports these measures to best reflect the environment of the child.

*Infant measures.* Infant Motor Development. We measure infant motor development with the Patterson Medical Peabody Developmental Motor Scales—Second Edition (PDMS-2) [77]. The PDMS-2 has sound psychometric properties and is used from birth to 5 years of age to assess gross and fine motor skills.

*Infant crying problems*: Frequency, intensity, and duration of infant cries are assessed via caregiver-report on a 14-item cry assessment [78].

*Infant sleep*: We assess infant sleep quality and patterns using the Brief Infant Sleep Questionnaire-Revised (BISQ-R), which is a reliable and valid parent-report questionnaire that assesses infants sleep onset time, sleep latency, and number of night-wakings [79]. It has been validated against actigraphy and sleep diaries and has high test-rest reliability [80].

*Infant feeding*: Caregivers fill out a 3-item questionnaire measuring breastfeeding practices since the birth [81]. We also use the Feeding Matters Infant and Child Feeding Questionnaire (ICFQ), an evidence-based tool which assesses children's feeding habits to identify feeding disorders. The ICFQ is psychometrically sound and has 6 core items that may be used to identify those infants that need pediatric feeding referrals [82].

*Infant temperament*: We assess infant temperament with the Infant Behavior Questionnaire-revised (IBQ) [83]. The IBQ has established psychometric support for measuring temperament related to activity level, smiling and laughter, distress and latency to novel stimuli, distress to limitations, soothability, and duration of orienting.

## 18 months (Time 3)

**Procedure.** For the final visit, the primary caregiver is asked to complete a series of questionnaires examining maternal sociodemographic risk, psychopathology, and trauma, as well as toddler language development, crying, sleeping, feeding, and temperament measures. We also obtain childcare information and medical history from the medical records to document the number and severity of medical illnesses. We use the Bayley Scales of Infant and Toddler Development IV to examine cognitive, language, and motor outcomes, and we measure toddler executive functioning skills with the Wand task, the A-not-B task, and the Hide the Pots task. Lastly, we collect a buccal sample from the toddler and collect anthropometrics (length, weight, head circumference).

**Maternal measures.** Maternal sociodemographic risk, psychopathology, and trauma surveys are repeated from the birth and 6-month visit (see Newborn and 6-month Maternal Measures). If the birthing parent is not available, the primary caregiver reports these measures to best reflect the environment of the child.

**Toddler measures.** *Neurodevelopmental outcomes*. We use the Bayley Scales of Infant and Toddler Development IV to assess cognitive, motor, and language outcomes [84]. The Bayley is an optimal neurodevelopmental assessment in early childhood given its strong reliability and moderately strong predictive validity [85].

*Toddler problem behavior*. We measure toddler problem behavior with the Infant and Toddler Social and Emotional Assessment (ITSEA) [86]. The ITSEA is a measure of internalizing and externalizing child behavior, as well as dysregulation and competence. It has test-retest and interrater reliability and strong construct validity in children age 12–36 months [87].

*Executive functioning*. We assess toddler executive functioning with the Glitter Wand task [88], the A-not-B task [89], and the Hide the Pots task [90]. All tasks measure key components of executive functioning including set shifting or shifting between rules, inhibitory control, and working memory.

*Expressive and receptive language*. We assess toddler language with the Bayley scales and maternal report of toddler receptive and expressive language development using the MacArthur Communicative Development Inventories [91] which has strong reliability and validity.

*Toddler feeding*. We measured toddler feeding problems with the ICFQ (For psychometrics see Infant Feeding under 6-month Infant Measures) for 15- to 24- month infants. For a full list of measures see Table 3.

*Infant temperament*. We assess infant temperament with the Infant Behavior Questionnaire-revised (IBQ) [83]. For more information see Infant Temperament in 6-month Infant Measures.

*Infant sleep*. We assess infant sleep quality and patterns using the Brief Infant Sleep Questionnaire-Revised (BISQ-R). For more information see Infant Sleep in 6-month Infant Measures.

**Table 3. Protocol measures.**

| Assessments | Tools | Time | | |
|---|---|---|---|---|
| | | **Birth** | **6m** | **18m** |
| *Maternal Sociodemographic* | Family Resource Scale | √ | √ | √ |
| | Demo; SES = income, education, and occupation | √ | √ | √ |
| *Maternal Psychopathology* | Patient Health Questionnaire–9 | √ | | |
| | Patient-Reported Outcomes and Measurement Information System | √ | √ | √ |
| | Borderline Symptoms Checklist | √ | √ | √ |
| | Difficulties in Emotion Regulation-Short Form | √ | √ | √ |
| | Connor-Davidson Resilience Scale | √ | | |
| *Maternal Trauma History* | Adverse Childhood Experiences Questionnaire | √ | | |
| | Benevolent Childhood Experiences Scale | √ | √ | √ |
| *NOWS Severity* | Medical Record Abstraction: Length of treatment, maximum dose, additional NOWS medications | | √ | √ |
| *Number of Caregiver Transitions* | Caregiver transitions | √ | √ | √ |
| *Infant Epigenetics* | Buccal | √ | √ | √ |
| *Opioid and Substance Exposure* | Hair toxicology | √ | | |
| | Substance Use Inventory | √ | √ | √ |
| *NOWS Diagnosis* | Medical Record Abstraction | √ | | |
| *Newborn birth characteristics* | Medical Record Abstraction | √ | | |
| *Newborn Neurobehavior* | NNNS-II | √ | | |
| *Cortisol* | Saliva | √x3 | | |
| *Infant Heart Rate* | Heart Rate Variability/ECG/Mindware | √x2 | | |
| *Infant Cry* | Cry recording | √ | | |
| | Cry Checklist | | √ | |
| *Infant Sleep* | EMFIT QS sleep sensor | √ | | |
| | Brief Infant Sleep Questionnaire–Revised | | √ | √ |
| *Infant Feeding* | Infant and Child Feeding Questionnaire | | √ | √ |
| | Infant breastfeeding questionnaire | | √ | |
| *Crying, Sleeping, & Feeding* | Follow-up & pediatric medical history interview | | √ | √ |
| *Neurodevelopmental Outcomes* | Peabody Developmental Motor Scales–2nd Ed. | | √ | |
| | Bayley Scales of Infant and Toddler Development–IV | | | √ |
| *Infant Temperament* | Infant Behavior Questionnaire | | √ | √ |
| *Toddler Problem Behavior* | Infant Toddler Social and Emotional Assessment | | | √ |
| *Expressive & Receptive Language* | MacArthur-Bates Communicative Developmental Inventories | | | √ |
| *Executive Functioning* | Wand Task | | | √ |
| | A not B Task | | | √ |
| | Hide the Pots Task | | | √ |

*Infant crying problems.* Frequency, intensity, and duration of infant cries are assessed via caregiver-report on a 14-item cry assessment [78].

## Data analyses

### Aim 1

**Assess infant neurobehavior, newborn autonomic biomarkers, prenatal opioid use, and sociodemographics as predictors of NOWS onset and severity.** We use generalized linear models to examine whether impaired newborn neurobehavior, abnormal cry, sleep, cortisol, HRV, epigenetics, prenatal opioid exposure as measured in maternal hair, and/or low socio-economic status are predictive of NOWS diagnosis and severity. We control for site, maternal

age, maternal psychopathology, gestational age, birth weight, newborn medical complications, and newborn sex. To control for several confounding variables, we use logistic regression to estimate the probability of NOWS diagnosis given the confounders; the resulting propensity scores are used as a covariate adjustment in subsequent analyses. As the occurrence of NOWS cannot be randomized, the use of propensity scores controls for imbalances across subject characteristics and allows for inferences to better approximate the findings of randomized clinical trials [92]. We also run models to determine if effects differ based on biological sex.

### Aim 2

**Evaluate the predictive validity of these clinical markers and test whether NOWS severity predicts neurodevelopment impairment at 6 and 18 months.** We use generalized linear models with a normally distributed outcome to examine impairment in neurodevelopment later in infancy and toddlerhood. Multiple regressions are run independently predicting social, emotional, developmental, and executive functioning outcomes at the 6- and 18-month assessments. The propensity scores from Aim 1 are utilized to control for confounding variables, and the number of caregiver transitions. We model main effects of the proposed predictors and interactions with NOWS diagnosis, and models to determine sex differences. The same consolidated measures are used from Aim 1, and subsequent exploratory analyses of consolidated measure components are evaluated using leave-one-out cross-validation.

### Aim 3

**Gauge the accuracy and feasibility of implementing proposed newborn autonomic biomarkers as NOWS diagnostic assessments.** We examine the diagnostic accuracy of our newborn autonomic biomarkers compared to the current diagnostic measures, the Finnegan, NWI, and ESC. We assess the sensitivity, specificity, and predictive value of each proposed newborn autonomic marker [93, 94] and potential combinations thereof.

## Discussion

We propose to advance NOWS standard-of-care by developing a diagnostic assessment based on replicable and reliable clinical markers indicative of infants' neurobehavioral outcomes. We aim to distinguish these early clinical markers of NOWS risk that are detectable in the first 24 hours after birth, using standardized tools that are routine care in some hospitals (NNNS-II), automated assessments that could be easily disseminated into well baby nurseries or NICUs (infant cry analyzer, sleep sensor), autonomic biomarkers (HRV, salivary cortisol, epigenetics), and maternal measures (hair toxicology, sociodemographic information).

This study protocol is not without limitations. Confounding factors like maternal stress—due to poverty, psychopathology, and poor general health, as well as exposure to violence, trauma, and/or abuse—may complicate the interpretations of our findings. For example, while fetal and newborn heart rate variability have been linked to opioid exposure [30, 31], greater endorsements of prenatal stress have also been associated with reduced fetal heart rate variability [95]. Similarly maternal stress may also play a role in infant cortisol levels [33, 36]. Therefore, we measure maternal sociodemographic risk, psychopathology, and trauma history in this protocol to account for potential confounding. Further, the *COMT* and *NR3C1* gene methylation are related to both infant stress and opioid exposure, which may confound findings. Site-level differences may also complicate findings; therefore, we account for site within each analysis. The current study protocol recruits from a hard to retain population, and the length of the study and the number of study measures increase the complexity of the study, which may make recruitment and retention less feasible. However, to combat this limitation,

we implement an evidence-based recruitment and retention plan for high-risk families (See Tables B & C in S1 Text).

Early identification of NOWS could lead to more precise treatment, which could reduce severity, contribute to more timely hospital discharges, decrease hospital costs, and improve care by eliminating disruptions inherent in a hospital setting. With this protocol, we could improve our capacity to care for this patient population.

## Supporting information

**S1 Text.**
(DOCX)

## Acknowledgments

We thank the families who continue to generously donate their time to participate in our study. Thank you to the dedicated staff research assistants.

## Author Contributions

**Conceptualization:** Sarah E. Maylott, Barry M. Lester, Sheila E. Crowell, Pascal Deboeck, Amy Salisbury, Elisabeth Conradt.

**Formal analysis:** Pascal Deboeck, Elisabeth Conradt.

**Funding acquisition:** Barry M. Lester, Elisabeth Conradt.

**Investigation:** Barry M. Lester, Elisabeth Conradt.

**Methodology:** Sarah E. Maylott, Barry M. Lester, Lydia Brown, Ayla J. Castano, Lynne Dansereau, Sheila E. Crowell, Pascal Deboeck, Amy Salisbury, Elisabeth Conradt.

**Project administration:** Barry M. Lester, Lydia Brown, Ayla J. Castano, Lynne Dansereau.

**Resources:** Barry M. Lester, Sheila E. Crowell, Amy Salisbury, Elisabeth Conradt.

**Supervision:** Barry M. Lester, Amy Salisbury, Elisabeth Conradt.

**Visualization:** Sarah E. Maylott.

**Writing – original draft:** Sarah E. Maylott.

**Writing – review & editing:** Sarah E. Maylott, Barry M. Lester, Lydia Brown, Ayla J. Castano, Lynne Dansereau, Sheila E. Crowell, Pascal Deboeck, Amy Salisbury, Elisabeth Conradt.

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
