## [Decision Letter · Decision Letter 0]

30 Oct 2023

PONE-D-23-04246Enhancing the diagnostic accuracy and predictive validity of neonatal opioid withdrawal syndrome: The utility of non-invasive clinical markersPLOS ONE

Dear Dr. Maylott,

Thank you for submitting your manuscript to PLOS ONE. After careful consideration, we feel that it has merit but does not fully meet PLOS ONE’s publication criteria as it currently stands. Therefore, we invite you to submit a revised version of the manuscript that addresses the points raised during the review process.

We look forward to receiving your revised manuscript.

Kind regards,

Ammal Mokhtar Metwally, Ph.D (MD)

Academic Editor

PLOS ONE

Journal Requirements:

2. Please include “Protocol” in the manuscript  title.

4. We note that you have referenced (Unpublished) om page 28, which has currently not yet been accepted for publication. Please remove this from your References and amend this to state in the body of your manuscript: (ie “Bewick et al. [Unpublished]”) as detailed online in our guide for authors

Additional Editor Comments:

The manuscript is interested meanwhile, the reviewers have raised a number of points which we believe would improve the manuscript and may allow a revised version to be published in PLOS one.

Reviewers' comments:

Reviewer's Responses to Questions

**Comments to the Author**

1. Does the manuscript provide a valid rationale for the proposed study, with clearly identified and justified research questions?

Reviewer #1: Yes

Reviewer #2: Yes

2. Is the protocol technically sound and planned in a manner that will lead to a meaningful outcome and allow testing the stated hypotheses?

Reviewer #1: Yes

Reviewer #2: Yes

3. Is the methodology feasible and described in sufficient detail to allow the work to be replicable?

Reviewer #1: Yes

Reviewer #2: Yes

4. Have the authors described where all data underlying the findings will be made available when the study is complete?

Reviewer #1: No

Reviewer #2: Yes

5. Is the manuscript presented in an intelligible fashion and written in standard English?

Reviewer #1: Yes

Reviewer #2: Yes

6. Review Comments to the Author

You may also provide optional suggestions and comments to authors that they might find helpful in planning their study.

Reviewer #1: On line 37, the word "consent" was left out and should be added.

Paragraph beginning line 79 - the authors should clarify that the mu-receptor binding, and then lack of, is primarily responsible for the symptoms of withdrawal as described; individuals not familiar with this area may not make that connection.

Line 95 - May want to change the wording "evidenced-based". Given the recent ongoing studies evaluating the assessment of infants with NOWS, this is now becoming evidenced-based. Would delete that phrase.

Line 106 authors state NNNS-II; later it is called NNNS. Please be consistent.

The authors report that infants exposed to both opioids and cocaine have louder and higher pitched cries (line 133). How can cry be used in infants with co-exposures? Is this a limitation?

The authors report that maternal stress can negatively impact fetal development, and thus impact the cortisol levels. How does the maternal stress play into the impact on HRV? That should be considered when assessing the HRV and may limit the applicability of HRV in certain populations. Similarly, including maternal stressors as a confounding factor in the cortisol measurements is important.

In assessing the COMT gene - has this also been assessed in individuals with higher stress levels to ensure that the increase in truly in relation to the opiate exposure?

The authors conclude in lines 217-220 that infants with a longer gestation have worse NOWS. This should be reconsidered. Currently, the assessments that are used are focused on the term infant and what is expected at that gestational age. This then leads to the inability to accurately assess preterm infants for NOWS, given that they have very different neurological development at birth and during the period of expected withdrawal. I would not conclude that term infants have worsex withdrawal, but rather that we are not able to assess withdrawal in preterm accurately.

Line 262 - states that data is collected 24 hours of life, when infant is pre-symptomatic. I would take out pre-symptomatic, as some infants do develop withdrawal symptoms at that time, especially when they are exposed to opioids with short half-lives.

What are "caregiver transitions"? This should be defined.

Does the removal of the HRV monitor result in stress in the infant that could impact the samples collected for cortisol / gene measurement?

Please include citations to support line 283 - the family resource scale has high test-retest reliability and predictive validity.

If the pregnant individual does not have long enough hair, are they excluded from the study?

Were all infants included in the study singleton? Term? Breastfed or bottle fed?

What is the timing of this study (later it is stated 2020)? Why was Bayley III used instead of IV? The Bayley III would have been outdated at that time.

The authors say "Infant measures" when describing the 18 month visit. The term infant is no longer applicable and should be changed.

If the cry analyzer is used in an open bay NICU, how accurate is it? Will it pick up the noise of surrounding infants?

Are there results of this protocol? There is a lot of description of the protocol, with almost no discussion and no results. The title implies that there should be results given. If this is meant to be a protocol paper, that should be clearly stated in the text rather than simply stated as the article type. The discussion should also include more points about confounding factors that may impact the assessments, strengths and weaknesses of the approach, etc.

Reviewer #2: I read with inerest the paper by Mylott and colleagues about a possible protocol for the early identification of NOWS, aiming to provide a more targeted approach to neonates ar risk, allowing an early identification of NOWS and also of those infants that will not develop this problem. moreover the later timepoints will allow to evaluate the neurodeveoplment of this particular population, that is evidently at risk for neurodevelopment impairment.

The suggested protocol enclosed multiple approaches to target from different sides the problem, analysing different aspects of the same issue.

The proposal is surely interesting and complete, and the results of its application will consequently be awaited.

The main concern about this approach is obviously the complexity, that requires many different approaches and is consequently time-consuming and makes it less feasible in a setting with less resources. Authors should adress this aspect.

7. PLOS authors have the option to publish the peer review history of their article (what does this mean?). If published, this will include your full peer review and any attached files.

Reviewer #1: No

Reviewer #2: No

---

## [Author Response · Author response to Decision Letter 0]

5 Jan 2024

Journal Requirements:

2. Please include “Protocol” in the manuscript title.

Authors: “Protocol” was added to the title: A protocol for enhancing the diagnostic accuracy and predictive validity of neonatal opioid withdrawal syndrome: The utility of non-invasive clinical markers

Authors: Because data collection is currently ongoing and no data was used in this protocol paper, no data is currently available to be shared. We have stated this in the Data Availability section: 

“Data Availability: As data collection for the present study protocol is ongoing, no study data is currently available. Deidentified research data will be made available upon reasonable request when the study is completed and published.”

4. We note that you have referenced (Unpublished) om page 28, which has currently not yet been accepted for publication. Please remove this from your References and amend this to state in the body of your manuscript: (ie “Bewick et al. [Unpublished]”) as detailed online in our guide for authors

 Authors: We added i.e., Weiss et al. [Unpublished] to the body of the text. 

Additional Editor Comments:

The manuscript is interested meanwhile, the reviewers have raised a number of points which we believe would improve the manuscript and may allow a revised version to be published in PLOS one.

Reviewers' comments:

Reviewer's Responses to Questions

Comments to the Author

1. Does the manuscript provide a valid rationale for the proposed study, with clearly identified and justified research questions?

Reviewer #1: Yes

Reviewer #2: Yes

2. Is the protocol technically sound and planned in a manner that will lead to a meaningful outcome and allow testing the stated hypotheses?

Reviewer #1: Yes

Reviewer #2: Yes

3. Is the methodology feasible and described in sufficient detail to allow the work to be replicable?

Reviewer #1: Yes

Reviewer #2: Yes

4. Have the authors described where all data underlying the findings will be made available when the study is complete?

Reviewer #1: No

Reviewer #2: Yes

5. Is the manuscript presented in an intelligible fashion and written in standard English?

Reviewer #1: Yes

Reviewer #2: Yes

6. Review Comments to the Author

You may also provide optional suggestions and comments to authors that they might find helpful in planning their study.

Reviewer #1: On line 37, the word "consent" was left out and should be added.

Authors: We thank the reviewer for pointing out this error. We added “consent” to line 37.

Paragraph beginning line 79 - the authors should clarify that the mu-receptor binding, and then lack of, is primarily responsible for the symptoms of withdrawal as described; individuals not familiar with this area may not make that connection.

Authors: We now clarify this connection for the reader: 

“Consequently, the μ-receptor binding of opioids and then the cessation binding (i.e., the cessation of opioid exposure) after birth may lead to withdrawal signs, including autonomic instability, high-pitched cry, irritability, tremors, and difficulty feeding and sleeping (12).”

Line 95 - May want to change the wording "evidenced-based". Given the recent ongoing studies evaluating the assessment of infants with NOWS, this is now becoming evidenced-based. Would delete that phrase.

Authors: We changed the wording “evidenced-based” on line 95 to describe the limitations of the current clinical assessments in more detail:

“However, these diagnostic tools are not often thoroughly researched, psychometrically sound, and/or standardized, and the choice of diagnostic tool depends on independent hospital protocols (15).”

Line 106 authors state NNNS-II; later it is called NNNS. Please be consistent.

Authors: We now refer to the NeoNatal Neurobehavioral Scale as the NNNS-II, based off the original NNNS exam.

The authors report that infants exposed to both opioids and cocaine have louder and higher pitched cries (line 133). How can cry be used in infants with co-exposures? Is this a limitation?

Authors: We discuss the implications of exposures and co-exposures for infant cry measures:

“Infants with in utero opioid exposure tend to have shorter cry utterances and extremely high-pitched cries, and infants co-exposed to opioids and cocaine have comparably louder and higher pitched cries (20). High-pitched and hyperphonated cries have been reported in infants with neurologic problems (21), suggesting that infants with substance exposures, especially those with polysubstance exposures may be at increased neurological risk.”

The authors report that maternal stress can negatively impact fetal development, and thus impact the cortisol levels. How does the maternal stress play into the impact on HRV? That should be considered when assessing the HRV and may limit the applicability of HRV in certain populations. 

Similarly, including maternal stressors as a confounding factor in the cortisol measurements is important.

Authors: We thank the reviewer for this feedback. We added a limitations section to discuss maternal stress as a confounding variable: 

“This study is not without limitations. Confounding factors like maternal stress—due to poverty, psychopathology, and poor general health, as well as exposure to violence, trauma, and/or abuse—may complicate the interpretations of our findings. For example, while fetal and newborn heart rate variability have been linked to opioid exposure (30,31), greater endorsements of prenatal stress have also been associated with reduced fetal heart rate variability (92). Similarly maternal stress may also play a role in infant cortisol levels (33,36). Therefore, we measure maternal sociodemographic risk, psychopathology, and trauma history in this protocol to account for potential confounding.”

In assessing the COMT gene - has this also been assessed in individuals with higher stress levels to ensure that the increase in truly in relation to the opiate exposure?

Authors: The COMT gene has been assessed in individuals with higher stress exposure. At present it is unclear whether associations between opioid exposure and COMT gene is due to opioid exposure or related exposures such as stress. We included this as a limitation: 

“Further, the COMT and NR3C1 gene methylation are also related to both infant stress and opioid exposure, which may confound findings.”

The authors conclude in lines 217-220 that infants with a longer gestation have worse NOWS. This should be reconsidered. Currently, the assessments that are used are focused on the term infant and what is expected at that gestational age. This then leads to the inability to accurately assess preterm infants for NOWS, given that they have very different neurological development at birth and during the period of expected withdrawal. I would not conclude that term infants have worse withdrawal, but rather that we are not able to assess withdrawal in preterm accurately.

Authors: We thank the reviewer for this suggestion. We added this an explanation for late term infants’ likelihood of being treated for NOWS: 

“These findings may be due to the high permeability of the placenta in the late third trimester (50) or that current assessments are not geared toward assessing NOWS in preterm infants who may display differences in neurological development and, therefore, withdrawal signs at birth.”

Line 262 - states that data is collected 24 hours of life, when infant is pre-symptomatic. I would take out pre-symptomatic, as some infants do develop withdrawal symptoms at that time, especially when they are exposed to opioids with short half-lives.

Authors: We thank the reviewer for highlighting this point. We removed “pre-symptomatic” as a descriptive as some of our infants may show symptoms prior to the 24-hour mark.

What are "caregiver transitions"? This should be defined.

Authors: We now define caregiver transitions:

“We record the number of primary caregiver changes over the course of the study, starting at birth. This occurs when there is a change in custody. The child’s primary caregiver at birth is the biological mother. Caregiver transitions occur when the child is placed with a foster parent, adoptive parent, or other caregiver (e.g., family member). Reasons for caregiver transitions include DCFS involvement (e.g., neglect, abuse, substance use), incarceration of caregiver, death of caregiver, adoption, and/or reunification, among others.”

Does the removal of the HRV monitor result in stress in the infant that could impact the samples collected for cortisol / gene measurement?

Authors: We now clarify that the removal of the HRV monitor does not impact the samples collected as we remove the ECG leads and wires after the saliva and buccal samples have been collected:

“ECG leads and wires are removed post newborn assessments, so that we cause no additional stress during the exams.”

Please include citations to support line 283 - the family resource scale has high test-retest reliability and predictive validity.

Authors: We thank the authors for pointing out this discrepancy. We report the citation to verify reliability and predictive validity of the Family Resource Scale 

61. Dunst CJ, Leet HE. Family Resource Scale. Child Care Health Dev [Internet]. 1987; Available from: https://doi.org/10.1037/t33262-000

If the pregnant individual does not have long enough hair, are they excluded from the study?

Authors: We retained all participants even if they were not able to provide hair samples:

“Finally, we collect maternal hair samples for toxicology analyses. If the hair is too short or birthing parents refuse to provide the hair sample, the missing data is tracked but participants are still included in the study.”

Were all infants included in the study singleton? Term? Breastfed or bottle fed?

Authors: Our sample is not limited to singleton term infants and infants can be breast or bottle fed. We clarify this in the study population section:

“Newborns are medically stable and include singleton and twin pregnancies, term and preterm infants, and breast- and bottle-fed infants.”

What is the timing of this study (later it is stated 2020)? Why was Bayley III used instead of IV? The Bayley III would have been outdated at that time.

Authors: We thank the reviewer for highlighting this error. The study officially began in 2021 and the Bayley IV is used. We updated the appropriate sections in the paper.

The authors say "Infant measures" when describing the 18 month visit. The term infant is no longer applicable and should be changed.

Authors: We now refer to “Infant measures” at 18-months as “Toddler measures.”

If the cry analyzer is used in an open bay NICU, how accurate is it? Will it pick up the noise of surrounding infants?

Authors: We collect cry data in open bay NICUs and Intermediate Care Nurseries. After recordings are uploaded, they are screened and filtered for ambient noise so that only the target infant cries are included in analyses.

Are there results of this protocol? There is a lot of description of the protocol, with almost no discussion and no results. The title implies that there should be results given. If this is meant to be a protocol paper, that should be clearly stated in the text rather than simply stated as the article type. 

Authors: We updated the title to include “protocol” to clarify there are no findings discussed within the manuscript. 

The discussion should also include more points about confounding factors that may impact the assessments, strengths and weaknesses of the approach, etc.

Authors: We thank the reviewer for the feedback. We added potential confounding factors and discussed their potential impact on the study (page 22 & 23).

Reviewer #2: I read with inerest the paper by Mylott and colleagues about a possible protocol for the early identification of NOWS, aiming to provide a more targeted approach to neonates ar risk, allowing an early identification of NOWS and also of those infants tha

---

## [Decision Letter · Decision Letter 1]

29 Feb 2024

PONE-D-23-04246R1A protocol for enhancing the diagnostic accuracy and predictive validity of neonatal opioid withdrawal syndrome: The utility of non-invasive clinical markersPLOS ONE

Dear Dr. Maylott,

Thank you for submitting your manuscript to PLOS ONE. After careful consideration, we feel that it has merit but does not fully meet PLOS ONE’s publication criteria as it currently stands. Therefore, we invite you to submit a revised version of the manuscript that addresses the points raised during the review process.

We look forward to receiving your revised manuscript.

Kind regards,

Ammal Mokhtar Metwally, Ph.D (MD)

Academic Editor

PLOS ONE

**Additional Editor Comments:**

The manuscript is interested meanwhile, the reviewers have raised a number of points which we believe would improve the manuscript and may allow a revised version to be published in PLOS one.

Reviewers' comments:

Reviewer's Responses to Questions

**Comments to the Author**

1. Does the manuscript provide a valid rationale for the proposed study, with clearly identified and justified research questions?

Reviewer #1: Yes

Reviewer #3: Yes

2. Is the protocol technically sound and planned in a manner that will lead to a meaningful outcome and allow testing the stated hypotheses?

Reviewer #1: Yes

Reviewer #3: Yes

3. Is the methodology feasible and described in sufficient detail to allow the work to be replicable?

Reviewer #1: Yes

Reviewer #3: Yes

4. Have the authors described where all data underlying the findings will be made available when the study is complete?

Reviewer #1: No

Reviewer #3: Yes

5. Is the manuscript presented in an intelligible fashion and written in standard English?

Reviewer #1: Yes

Reviewer #3: Yes

6. Review Comments to the Author

You may also provide optional suggestions and comments to authors that they might find helpful in planning their study.

Reviewer #1: Under clinical measure 2: citation should be added to line 138-139: "preliminary acoustic cry data showed that it was 81% accurate in predicting NOWS."

Lines 150-151: While the higher doses of opioid exposure show more disturbance in sleep, higher doses of opioid exposure are not necessarily linked to more severe withdrawal in the infant. Please revise this sentence.

Please revise lines 221-224. Why would an infant with a longer gestation spend longer time in the hospital compared to an infant born at a shorter gestation (ie - 23 weeks gestation). Also, the correlation between dose exposure and severity of withdrawal continues to be debated.

Methods - when is the cry recording started? The methods states it continues during the NNNS-II but doesn't say the start. Does the infant keep the HRV leads on during the NNNS-II assessment? Is this felt to have any impact on the results of the NNNS?

Was IRB approval obtained - this was found later in the manuscript at the end of the methods? At one point consented is mentioned, but IRB approval is not included. Please move the IRB information, with the date range of enrollment (not given) to the beginning of the methods.

Line 297 - "caregiver changes may be an indicator of problem behavior." Do you have evidence to support this is true in this population? This is a high risk population, and so caregiver changes may occur due to caregiver indications. Consider removing this sentence.

What was the timing of enrollment in the study? I ask given that Finnegan was used to assess withdrawal, and ESC is becoming the gold standard approach for assessment.

If the caregiver at the 6 month visit was not the birthing caregiver, what data was collected (if any)?

Consider changing maternal measures to birthing caregiver measures

Under the 6 month infant measures, please remove those items collected at 18 months, at that makes it confusing to readers. Include those items under the 18 month infant measures.

If the caregiver at the 18 month visit was not the birthing caregiver, what data was collected (if any)?

Consider changing maternal measures to birthing caregiver measures

Reviewer #3: It is a carefully done piece of work. I am extremely satisfied with the content, structure, and scientific approach of this study protocol. I firmly believe that it is a great work with great public health significance.

7. PLOS authors have the option to publish the peer review history of their article (what does this mean?). If published, this will include your full peer review and any attached files.

Reviewer #1: No

Reviewer #3: No

---

## [Author Response · Author response to Decision Letter 1]

30 Apr 2024

Comments to the Author

1. Does the manuscript provide a valid rationale for the proposed study, with clearly identified and justified research questions?

Reviewer #1: Yes

Reviewer #3: Yes

2. Is the protocol technically sound and planned in a manner that will lead to a meaningful outcome and allow testing the stated hypotheses?

Reviewer #1: Yes

Reviewer #3: Yes

3. Is the methodology feasible and described in sufficient detail to allow the work to be replicable?

Reviewer #1: Yes

Reviewer #3: Yes

4. Have the authors described where all data underlying the findings will be made available when the study is complete?

Reviewer #1: No

Reviewer #3: Yes

5. Is the manuscript presented in an intelligible fashion and written in standard English?

Reviewer #1: Yes

Reviewer #3: Yes

6. Review Comments to the Author

You may also provide optional suggestions and comments to authors that they might find helpful in planning their study.

Reviewer #1: 

Under clinical measure 2: citation should be added to line 138-139: "preliminary acoustic cry data showed that it was 81% accurate in predicting NOWS."

Authors: We thank the reviewer for pointing out this discrepancy. We reference preliminary data that is unpublished, so we added “(Unpublished data)” at the conclusion of the sentence: 

“Preliminary acoustic cry data showed that it was 81% accurate in predicting NOWS; moreover, acoustics were more accurate in some cases when the observational crying score was incorrectly evaluated (Unpublished data).”

Lines 150-151: While the higher doses of opioid exposure show more disturbance in sleep, higher doses of opioid exposure are not necessarily linked to more severe withdrawal in the infant. Please revise this sentence.

Authors: We appreciate the feedback and agree that higher doses of opioid exposure do not necessarily result in more severe withdrawal; therefore, we changed “severity” to “neurobehavior” as outlined below:

“Further, newborns exposed to higher doses of opioids in utero showed more disturbances in neonatal sleep, indicating that sleep patterns provide insight into the neurobehavior of newborn withdrawal (26).”

Please revise lines 221-224. Why would an infant with a longer gestation spend longer time in the hospital compared to an infant born at a shorter gestation (ie - 23 weeks gestation). Also, the correlation between dose exposure and severity of withdrawal continues to be debated.

Authors: We agree that drawing conclusions from this literature may be difficult; however, we feel that it is important information to outline in the background. We have taken steps to clarify that hospital stay was associated with higher doses of opioid treatment, but that length of NOWS treatment was associated with gestational age. In other words, a 23-week preterm infant may be in the hospital longer, but may undergo treatment for NOWS for a shorter period than infants with longer gestations. We provide a theory on why this may be occurring. We also acknowledge that dose and severity of withdrawal is a topic that continues to be debated.

“Newborns exposed to higher doses of methadone in utero were more likely to be treated for NOWS and to spend longer in the hospital than newborns exposed to lower doses (50); however, the effect of opioid dose on the severity of withdrawal continues to be debated. Further, newborns were treated for NOWS longer if they had a longer gestation (37-42 weeks treated for an average of 39.5 days; 33-36 weeks treated for an average of 20 days; 23-32 weeks treated for an average of 10 days). These findings may be due to the high permeability of the placenta in the late third trimester (51) or that current assessments are not geared toward assessing NOWS in preterm infants who may display differences in neurological development and, therefore, withdrawal signs at birth.”

Methods - when is the cry recording started? The methods states it continues during the NNNS-II but doesn't say the start. Does the infant keep the HRV leads on during the NNNS-II assessment? Is this felt to have any impact on the results of the NNNS?

Authors: The cry recording starts at the beginning of the NNNS-II assessment. We stop the recording at the end of the NNNS-II and then restart the cry recorder. We leave the cry recorder on for the remaining duration of the infant’s hospital stay. This results in distinct recordings of the infant’s cry during the NNNS-II and after the NNNS-II. 

“The cry recording begins during the NNNS-II, is stopped after the assessment concludes, and then is restarted to record continuously for the length of hospital stay.”

The HRV leads are left on the infant’s chest during the NNNS-II assessment to avoid additional stress to the infant prior to the NNNS-II. The HRV leads are not felt to have any impact on the results of the NNNS-II.

“Then we measure infant heartrate, placing three electrodes on the upper (negative) and lower (positive) chest around the infant’s heart, and on the right lower side of the chest (neutral) and are left on the infant for the full newborn visit.”

Was IRB approval obtained - this was found later in the manuscript at the end of the methods? At one point consented is mentioned, but IRB approval is not included. Please move the IRB information, with the date range of enrollment (not given) to the beginning of the methods.

Authors: We moved the IRB information to the beginning of the Methods sections and included the dates of enrollment/data collection.

“This study was approved by the Institutional Review Board at the University of Utah on September 14th, 2019 (00124221) and Women and Infants Hospital on February 13th, 2020 (1479081-3). Written consent was obtained for all participants. Enrollment and data collection began in 2019 and will be completed in 2025.”

Line 297 - "caregiver changes may be an indicator of problem behavior." Do you have evidence to support this is true in this population? This is a high risk population, and so caregiver changes may occur due to caregiver indications. Consider removing this sentence.

Authors: We thank the reviewer for the feedback. Given the robust evidence that caregiver changes are related to increased risk for problems behavior, we theorized that caregiver changes may have a detrimental impact on development. We now cite this literature.

Fisher PA, Stoolmiller M, Mannering AM, Takahashi A, Chamberlain P. Foster placement disruptions associated with problem behavior: Mitigating a threshold effect. J Consult Clin Psychol. 2011;79(4):481–7. 

Bada HS, Langer J, Twomey J, Bursi C, Lagasse L, Bauer CR, et al. Importance of Stability of Early Living Arrangements on Behavior Outcomes of Children With and Without Prenatal Drug Exposure. J Dev Behav Pediatr [Internet]. 2008 Jun [cited 2024 Apr 14];29(3):173. Available from: https://journals.lww.com/jrnldbp/fulltext/2008/06000/importance_of_stability_of_early_living.5.aspx?casa_token=686Fig2bOS4AAAAA:aVWtBIgzo_M0AtjbgFr_b8

What was the timing of enrollment in the study? I ask given that Finnegan was used to assess withdrawal, and ESC is becoming the gold standard approach for assessment.

Authors: The enrollment of the study began in 2019. Some participants are assessed with the Eat, Sleep, Console (ESC), as the University of Utah Hospital implemented this assessment in the maternal newborn unit in 2023; however, the NWI is still used in other units (e.g., NICU). We now cite and report use of the ESC.

If the caregiver at the 6 month visit was not the birthing caregiver, what data was collected (if any)? Consider changing maternal measures to birthing caregiver measures.

Authors: We appreciate the reviewer for highlighting this point. If the caregiver at the 6-month visit is not the birthing parent, we collect all the same self-report measures from the primary caregiver; however, they complete a different version of the infant feeding survey (Infant Feeding CG), which has modified questions regarding breastfeeding.

“If the birthing parent is not available, the primary caregiver reports these measures to best reflect the environment of the child.”

Under the 6 month infant measures, please remove those items collected at 18 months, at that makes it confusing to readers. Include those items under the 18 month infant measures.

Authors: We apologize for the confusion. We assessed several measures at both 6 and 18 months, we know provide in depth information for each measure at the first age it is used and then list the measure at the remaining ages it is used, while referencing the previous section.

If the caregiver at the 18 month visit was not the birthing caregiver, what data was collected (if any)? Consider changing maternal measures to birthing caregiver measures

If the caregiver at the 18-month visit is not the birthing parent, we collect all the same self-report measures from the primary caregiver. 

“If the birthing parent is not available, the primary caregiver reports these measures to best reflect the environment of the child.”

Reviewer #3: It is a carefully done piece of work. I am extremely satisfied with the content, structure, and scientific approach of this study protocol. I firmly believe that it is a great work with great public health significance.

Authors: We appreciate the reviewer’s time and the positive feedback.

7. PLOS authors have the option to publish the peer review history of their article (what does this mean?). If published, this will include your full peer review and any attached files.

Do you want your identity to be public for this peer review? For information about this choice, including consent withdrawal, please see our Privacy Policy.

Reviewer #1: No

Reviewer #3: No

---

## [Decision Letter · Decision Letter 2]

12 Jun 2024

A protocol for enhancing the diagnostic accuracy and predictive validity of neonatal opioid withdrawal syndrome: The utility of non-invasive clinical markers

PONE-D-23-04246R2

Dear Dr. Maylott,

We’re pleased to inform you that your manuscript has been judged scientifically suitable for publication and will be formally accepted for publication once it meets all outstanding technical requirements.

Kind regards,

Ammal Mokhtar Metwally, Ph.D (MD)

Academic Editor

PLOS ONE

Additional Editor Comments (optional):

Reviewers' comments:

Reviewer's Responses to Questions

**Comments to the Author**

1. Does the manuscript provide a valid rationale for the proposed study, with clearly identified and justified research questions?

Reviewer #1: Yes

Reviewer #3: Yes

2. Is the protocol technically sound and planned in a manner that will lead to a meaningful outcome and allow testing the stated hypotheses?

Reviewer #1: Yes

Reviewer #3: Yes

3. Is the methodology feasible and described in sufficient detail to allow the work to be replicable?

Reviewer #1: Yes

Reviewer #3: Yes

4. Have the authors described where all data underlying the findings will be made available when the study is complete?

Reviewer #1: Yes

Reviewer #3: Yes

5. Is the manuscript presented in an intelligible fashion and written in standard English?

Reviewer #1: Yes

Reviewer #3: Yes

6. Review Comments to the Author

You may also provide optional suggestions and comments to authors that they might find helpful in planning their study.

Reviewer #1: I appreciate the responsiveness by the authors and the changes made to the manuscript. No further revisions required at this time.

Reviewer #3: Well-designed, analyzed and presented study with a great positive impact for scientific world. It is high quality study.

7. PLOS authors have the option to publish the peer review history of their article (what does this mean?). If published, this will include your full peer review and any attached files.

Reviewer #1: **Yes: **Jessie Maxwell, MD, MBA

Reviewer #3: No

---

## [Editor Report · Acceptance letter]

17 Jul 2024

PONE-D-23-04246R2 

PLOS ONE

Dear Dr. Maylott, 

I'm pleased to inform you that your manuscript has been deemed suitable for publication in PLOS ONE. Congratulations! Your manuscript is now being handed over to our production team.

Kind regards, 

on behalf of

Professor Ammal Mokhtar Metwally 

Academic Editor

PLOS ONE